# A Transparency Sheet-Based Colorimetric Device for Simple Determination of Calcium Ions Using Induced Aggregation of Modified Gold Nanoparticles

**DOI:** 10.3390/ijms20122954

**Published:** 2019-06-17

**Authors:** Paweenar Duenchay, Orawon Chailapakul, Weena Siangproh

**Affiliations:** 1Department of Chemistry, Faculty of Science, Srinakharinwirot University, Sukhumvit 23, Wattana, Bangkok 10110, Thailand; paweenar13@gmail.com; 2Electrochemistry and Optical Spectroscopy Centre of Excellence (EOSCE), Department of Chemistry, Faculty of Science, Chulalongkorn University, Patumwan, Bangkok 10330, Thailand; corawon@chula.ac.th

**Keywords:** gold nanoparticles, modified gold nanoparticles, colorimetric analysis, calcium, transparency sheet-based device

## Abstract

A simple and novel transparency sheet-based colorimetric detection device using gold nanoparticles (AuNPs) modified by 4-Amino-6-hydroxy-2-mercaptopyrimidine monohydrate (AHMP) was fabricated and developed for the determination of calcium ions (Ca^2+^). The detection was based on a colorimetric reaction as a result of the aggregation of modified AuNPs induced by Ca^2+^ due to the ability to form strong electrostatic interactions between positively charged Ca^2+^ and negatively charged modified AuNPs. Probe solution changes color from red to blue in the presence of Ca^2+^ and can be observed by the naked eyes. To verify the complete self-assembly of the AHMP onto the AuNP surface, the modified AuNPs were characterized using ultraviolet–visible spectroscopy and zeta potential measurements. Under optimal conditions, a quantitative linearity was 10 to 100 ppm (*R*^2^ = 0.9877) with a detection limit of 3.05 ppm. The results obtained by the developed method were in good agreement with standard atomic absorption spectrometry (AAS) results and demonstrated that this method could reliably measure Ca^2+^. Overall, this novel alternative approach presents a low-cost, simple, sensitive, rapid, and promising device for the detection of Ca^2+^.

## 1. Introduction

Calcium (Ca^2+^) is an essential nutrient and one of the most important minerals for the human body. Ca^2+^ plays a role in the growth and maintenance of bones, teeth, and nails. In the ionized form, Ca^2+^ is significant in many vital processes, such as hormone secretion, nerve conduction, and blood coagulation. The World Health Organization (WHO) and Food and Agriculture Organization (FAO) recommend a minimum daily intake of 1200 mg of Ca^2+^ for adults [1,2]. At the present, calcium deficiencies are taken quite seriously. Therefore, in medical diagnostics, the determination of calcium concentration is very important for the evaluation of a patient’s health. A change in Ca^2+^ concentration, which usually is relatively stable in body fluids (range in urine: 100 to 300 mg/day) [3], can indicate the presence of various pathological conditions. When calcium concentrations are lower than normal, osteoporosis, vitamin D deficiency, eclampsia, or hypoparathyroidism may be present, whereas calcium concentrations higher than normal may indicate hyperparathyroidism, vitamin D intoxication, or myeloma [4]. Thus, Ca^2+^ concentration from body fluids should be determined for the evaluation of and protection against any disease that would be manifested by an abnormal calcium concentration. This emphasizes the need to develop highly sensitive devices for the portable, simple, and rapid deficiency screening in patient care. Numerous methods have been proposed for the measurement of Ca^2+^ in biological materials. The classical method involves titration with a standardized solution of ethylenediaminetetraacetic acid (EDTA) [5,6]. However, this method is limited by sensitivity and selectivity. The preferred standard method for the determination of Ca^2+^ in body fluids is atomic absorption spectrometry (AAS) [7,8,9,10]. Other spectrophotometries, such as UV–Vis spectroscopy [11], spectrofluorimetry [12], atomic emission spectrometry (AES) [13], inductively coupled plasma [14], and fluorescence [15,16,17] have also been developed for Ca^2+^ determination. Currently, the determination of Ca^2+^ developed with colored complexes using metallochromic reagents, such as Eriochrome Black T (EBT), has been reported [18,19]. Even though these spectrophotometric methods provide enough sensitivity, they are difficult to design to be a portable device. Calcium ion-selective electrodes are some of the most promising methods for Ca^2+^ determination. Given their precise and selective performance, this electrochemical measurement technique has been widely developed for the detection of Ca^2+^ in biological fluids [20,21,22]. There have been several kinds of electrodes developed for Ca^2+^ detection, including solid-contact ion selective electrodes [23] and polymer membrane ion-selective electrodes [24]. Unfortunately, most of the above-mentioned methods still require a specialist and specific instrument. Presently, with the rapid growth of near-patient devices used at home or the hospital bedside, there is an increasing demand for portable systems utilizing small disposable sensors which are capable of measuring biomarkers in biological samples. Patients can collect biological samples and measure target analyst concentrations at home within a few minutes without specialized instruments. From this requirement, it is the purpose of this work to present a simple, rapid, reliable, and low-cost detection method that uses a small sample volume for the determination of Ca^2+^ in urine samples using colorimetric gold nanoparticles. Recently, paper-based analytical devices (PADs) are one of the most common devices used as a miniaturized analytical platform for economical and portable sensing applications. Paper-based platforms can be beneficial for applications in diagnostics, food safety, and environmental monitoring due to the advantages of natural abundance, simplicity, disposability, low sample-reagent usage, and suitability for routine measurement. Unfortunately, the use of paper as substrate for colorimetric detection is usually suffer from the adsorption of color by porosities of paper lead to difficulty in the measurement of the color change at low concentration of the analyte. Therefore, the sensitivity in detection is low in comparison to those obtained from solution. To overcome this limitation, the search for new materials to be used as a substrate to fabricate the analytical platform is a challenge. In this research, we propose to use transparency sheets as the substrate for creating a new colorimetric device platform. 

Gold nanoparticles (AuNPs) have been widely used as colorimetric probes for metal ions, anions, small molecules, proteins, nucleic acids, and other analytes because of their unique properties, such as optical properties dependent on their spacing or surface plasmon resonances. The extinction spectrum and the wavelength at which AuNPs absorb and scatter light depends on the distance between particles [25,26]. For example, citrate-coated AuNPs were employed as colorimetric probes for organophosphorus pesticides detection [27]. Furthermore, AuNPs are easily modified by capping agents, which are specific to a given analyte. Therefore, functionalized AuNPs have been applied as colorimetric probes, receiving great attention in visual sensing applications because of the plasmonic absorbance shift. The functionalization can perform using various ligand; AuNPs functionalized with peptide substrates (Gly-Pro-Asp-Cys (GPDC) or Val-Pro-ethylene diamine-Asp-Cys (VP-ED-DC)) were proposed for the determination of dipeptidyl peptidase IV activity (DPP-IV/CD26 activity) based on the aggregation of functionalized AuNPs [28]. 4-Amino-6-hydroxy-2-mercaptopyrimidine monohydrate (AHMP) is an interesting ligand. It contains a thiol group which can self-assemble on the AuNP surface [29,30]. Additionally, the amine and hydroxyl groups available in AHMP are free from binding. The presence of the lone pair electrons of amine and hydroxyl groups lead to a negative charge around the AuNP surface. After modification, AHMP-AuNPs have a different zeta potential from the unmodified, indicating a negative charge. From this finding, it prompts us to use AHMP-AuNPs as the probes for Ca^2+^ detection. The electrostatic force between the negatively charged AHMP-AuNPs and positively charged Ca^2+^ may induce aggregation. The color change of AHMP-AuNPs has become a useful tool in the development of colorimetric sensing, transparency sheet-based analytical devices. These devices can determine the Ca^2+^ concentration in urine samples by assessment with the naked eye. The proposed mechanism for Ca^2+^ detection, the induction of the aggregation of AHMP-AuNPs, is shown in Scheme 1. AHMP molecules can be absorbed onto the surface of the AuNPs via the thiol group, resulting in the enhancement of the negative charge around the surface of the AuNPs after modification. Ca^2+^ can induce the aggregation of AHMP-modified AuNPs via electrostatic force. The red color of the AHMP-AuNPs changes to blue in accordance with the Ca^2+^ level added. Moreover, the proposed approach has been successfully applied to detect Ca^2+^ levels in urine samples. The verified results were in good agreement with atomic absorption spectroscopy (AAS). The simplicity, speed, suitability for real-time on-site detection, and lack of required complex and expensive instruments could make this method a promising tool for point-of-care (POC) diagnosis. 

## 2. Results and Discussion

### 2.1. Characterization of AHMP-AuNPs 

To investigate the incorporation of AHMP onto the AuNP surface, UV–Vis spectroscopy and zeta potentials were determined. Figure 1 shows the UV–Vis spectra of AHMP, AuNPs, and AHMP-AuNPs. The localized surface plasmon resonance (LSPR) absorption peak of AHMP has a maximum peak at 275 nm. The AuNPs alone have a maximum peak at 519 nm. After the AuNPs were modified with AHMP, the UV–Vis spectra showed two maximum peaks at 273 and 523 nm. We believe that the additional peak and the red shifting of the band is mainly caused by the decrease of the plasma oscillation frequency around the nanoparticles from the binding between thiol-containing compounds and the gold nanoparticles. Corresponding to the zeta potential results, the surface charge of AuNPs and AHMP-AuNPs were found at −43.1 and −49.9 mV, respectively. The zeta potentials indicated the negative charge of AuNPs increased from −43.1 mV (intensity 32 a.u.) to −49.9 mV (intensity 38 a.u.), as shown in Appendix A. This confirms that the surface of the AHMP-AuNPs has a more negative charge which corresponds to previous work. It is expected that the thiol group of AHMP is chemisorbed on the surface of AuNPs, whereas amine and hydroxyl groups are available in AHMP free from binding. The presence of lone pair electrons in amine and hydroxyl groups led to the negative charge around the AuNP surface. This result displayed that AHMP had been completely modified onto the surface of AuNPs.

### 2.2. Colorimetric Assay of Ca^2+^

Following the AuNP modification, a colorimetric analysis using AHMP-AuNPs was performed for the detection of Ca^2+^. The color and absorption spectra of the AHMP-AuNPs, with and without Ca^2+^, are shown in Figure 2, while their zeta potentials were displayed in Appendix A. After addition of buffer, the negative charge of AHMP-AuNPs solution increased from −49.9 mV to −58.2 mV, while the average size of the buffer-AHMP-AuNPs was not significantly change (from 29.1 to 27.6 nm). After addition of Ca^2+^, the zeta potential dramatically decreased from −58.2 mv, intensity 35 a.u., to −50.3 mv, intensity 26 a.u., while the average size of the AHMP-AuNPs increased from 29.1 to 65.4 nm. These results indicate that the AHMP-AuNPs aggregation was induced by the electrostatic force between the positive charge of Ca^2+^ and the negative charge of AHMP-AuNPs. Moreover, the results obtained corresponded to the decrease of the absorption bands at 273 and 523 nm along with the formation of a new absorption band at 680 nm (redshift), as illustrated in Figure 2. Furthermore, the color of the AHMP-AuNPs on the proposed device was clearly observed to change from red to blue immediately, which suggested the growth of the nanoparticles’ size. This indicates that Ca^2+^ could quickly bind to their functional groups (–NH_2_ and –OH) onto modified AuNPs [31,32]. These binding interactions and electrostatic force all contribute to the aggregation mechanism of the AuNPs, as schematically illustrated in Scheme 1. According to the considerable change in negative charge, color, and LSPR absorption of AHMP-AuNPs upon the addition of Ca^2+^, the development of a novel colorimetric AHMP-AuNPs device for Ca^2+^ determination was successfully accomplished. 

### 2.3. Optimization for the Determination of Ca^2+^ on Transparency Sheet-Based Devices

In this section, the modifier volume, pH, incubation time, and AHMP-AuNP: sample volume ratios were examined for optimization in order to accelerate the performance of the transparency sheet-based colorimetric device. The colorimetric detection of Ca^2+^ was performed at room temperature by the sequential addition of standard Ca^2+^ and AHMP-AuNPs on the detection zone of the transparency sheet. The red/blue intensity was measured and calculated by mobile phone and Image J software, respectively. The result of the aggregation capability between AHMP-AuNPs and Ca^2+^ in each condition was observed by the red/blue intensity from each transparency sheet image. First, the volume of 4 mM AHMP modifier was varied at 0.1, 0.2, 0.4, 0.6, 0.8, 1.0, 1.2, and 1.5 mL with the same volume of concentrated AuNPs (1.5 µL). Then, 15 µL of the varied AHMP-AuNPs were investigated with 15 µL of Ca^2+^ at 0, 10, 50, and 100 ppm concentration. In Figure 3, the red/blue intensity of the blank (AHMP-AuNPs and PBS buffer, pH 6) decreased when the AHMP volumes were increased. It may be that the AuNPs were diluted by the AHMP solution, so that AHMP-AuNPs had reduced aggregation efficiency. The red/blue intensity of 0.1 mL AHMP was found to exhibit the most significant change from the blank, especially when tested with 100 ppm Ca^2+^. Moreover, this change could be easily detected with the naked eye. Therefore, 0.1 mL AHMP was selected as an optimum modifier volume for further studies.

Then, the effect of pH (range of 3–12) on the aggregation process was studied. As shown in Figure 4, the use of phosphate buffer with pH of 5.8–12 did not change the color of the blank. Therefore, phosphate buffers of pH 5.8–12 were chosen to study the aggregation effect of AHMP-AuNPs and 100 ppm Ca^2+^ at a 1:1 volume ratio. The results demonstrated that pH 6 provided the maximum color change. At this pH, Ca^2+^ has a high potential to induce the aggregation of AHMP-AuNPs. Therefore, phosphate buffer pH 6 was chosen for further studies.

Next, the aggregation of AHMP-AuNPs induced by Ca^2+^ was investigated at various incubation times (1–30 min). The aggregation of the AHMP-AuNPs occurred immediately after the addition of Ca^2+^ and was completed in one minute. In other words, the red/blue intensity from the aggregation of AHMP-AuNPs induced by Ca^2+^ can be measured at one minute. When AHMP-AuNPs were left on the device for over five minutes, the red/blue intensity of the blank was increased from the initial (Appendix A) because of self-aggregation of the AHMP-AuNPs. In the case of interferences, Mg^2+^ was first selected as a representative of other positive ions, as it has an atomic radius and positive charge nearly equal to those of Ca^2+^ to investigate the possibility of using this proposed probe. From these results, it was found that the red/blue intensity of AHMP-AuNPs in the presence of Mg^2+^ was decreased more than 5%, compared to the blank, after five minutes. This indicated that other positive ions may also interfere with the proposed method if the incubation time was over five minutes. From this critical observation, we strongly believe that an incubation time of one minute is appropriate to specify the detection of Ca^2+^ by electrostatic force and size selective in the presence of other divalent ions. Therefore, the incubation time is a critical point that must be kept in mind for the selectivity study in the following section. Finally, to find a suitable quantity of the AHMP-AuNPs to sufficiently interact with Ca^2+^, the reagent volume ratio was tested between 5:25 and 25:5 (AHMP-AuNPs:Ca^2+^). The red/blue intensity of AHMP-AuNPs after mixing with Ca^2+^ was measured at various reagent volume ratios with 30 µL maximum volume. As shown in Appendix A, the volume ratio of AHMP-AuNPs:Ca^2+^ at 25:5 provided the highest red/blue intensity. This indicated that a 25:5 ratio of AHMP-AuNPs:Ca^2+^ was sufficient and appropriate for the determination of Ca^2+^.

### 2.4. Selectivity for the Determination of Ca^2+^

After getting the optimal conditions, selectivity was evaluated by the Δred/blue intensity (red/blue intensity of blank—red/blue intensity of sample) of AHMP-AuNPs mixed with other substances and metal ions possibly found in urine (i.e., ascorbic acid, uric acid, albumin, glucose, Mg^2+^, K^+^, Na^+^, Cl^−^, CO_3_^−^, PO_4_^2−^, and SO_4_^2−^), since they may also affect the aggregation of AHMP-AuNPs and might interfere with the method accuracy. An interference was defined to happen when the Δred/blue intensity in the mixture varied by more than ±5% of the Ca^2+^ intensity. As emphasized previously, the incubation time directly affects the selectivity. Therefore, the change of color was measured at one minute after mixing to neglect the interference from Mg^2+^. As shown in Figure 5, the results demonstrated clearly that the Δred/blue intensity of AHMP-AuNPs mixed with foreign compounds with concentrations even 10 times greater than the concentration of Ca^2+^ did not significantly affect the signal of Ca^2+^ and the tolerance limits were shown in Appendix A. In addition, we also applied this probe for detection of other divalent cations such as Hg^2+^, Cu^2+^, Cd^2+^, and Zn^2+^. According to the results, it can be seen that the interferences from almost all metal ions studied did not affect Ca^2+^ detection. Therefore, this probe provides high selectivity for only Ca^2+^ detection at an incubation time of one minute.

### 2.5. Analytical Performances

Under the optimal conditions, the analytical performance for determination of Ca^2+^ was investigated by AHMP-AuNPs using transparency sheet-base devices. Ca^2+^ concentrations were studied between 0 ppm and 100 ppm. As shown in Figure 6, it was found that the Δred/blue intensity from the aggregation of AHMP-AuNPs by Ca^2+^ was significantly reduced when the concentration of Ca^2+^ was increased. A linear relationship between the Δred/blue intensity and log concentration of Ca^2+^ in a calibration curve was obtained. A good linearity was observed for Ca^2+^ concentrations ranging from 10 ppm to 100 ppm with a correlation coefficient of 0.9877. The detection limit from calculation by using three times the standard deviation of the blank divided by the slope of the calibration curve (3 S_blank_/slope) was found to be 3.05 ppm. The proposed method provided a good recovery in the range of 91.7%–106.7%. These recoveries were within the acceptable recovery percentages, 90%–107%, for a concentration of 100 ppm (mg/kg) or at a Ca^2+^ concentration lower than 100 ppm, as recommended by the Association of Official Analytical Chemists (AOAC). The repeatability of this probe was investigated using 40 and 80 ppm standard solutions of Ca^2+^ (*n* = 5 measurements with different probes). The relative standard deviations were 2.54% and 5.16%, which indicated that there was good probe-to-probe reproducibility.

The comparisons between some previous report for the determination of Ca^2+^ and proposed devices were summarized in Appendix A. This proposed device demonstrates the ability of a transparency sheet-based colorimetric detection system to conveniently measure Ca^2+^, even though this proposed method has a detection limit higher than previous works, as shown in Appendix A. However, this limit of detection can be accepted for the determination of Ca^2+^ in urine samples. Moreover, this proposed platform offers the benefit in terms of simplicity, speed, suitability for real-time on-site detection, and lack of required complex and expensive instruments. Therefore, it can provide a promising tool for Ca^2+^ diagnosis and could be an alternative choice for point-of-care analysis in the future. 

### 2.6. Analysis of Artificial Urine Samples and Method Validation

To evaluate the efficiency of the proposed method, the transparency sheet-based devices were used to detect Ca^2+^ in artificial urine samples, and the results were compared to the AAS method as shown in Appendix A. The urine samples were spiked with standard Ca^2+^ at 80, 160, and 320 ppm. Then, the samples were further diluted to produce a final appropriate dilution factor before analysis. All artificial urine samples were analyzed by the AAS standard method and the proposed method. The obtained concentrations from the two methods were statistically compared using the *t*-test at the 95% confidence level. A *t*-test result was obtained with a value of 1.75, which is lower than the *t*_critical_ of 4.30. Thus, there was no significant difference between the concentrations obtained from these two methods. As a result, this proposed colorimetric method can be applied for the determination of Ca^2+^ in urine samples with satisfactory results, which suggests that the transparency sheet-based colorimetric device is accurate and reliable for practical applications in the clinical field.

## 3. Experimental

### 3.1. Materials and Instrumentation

Gold nanoparticles (AuNPs, 0.01% *w*/*v* (100 ppm), diameter 20 nm) were purchased from Kestrel Bio Sciences (Bangkok, Thailand). Calcium carbonate (CaCO_3_), sodium chloride (NaCl), magnesium chloride (MgCl_2_), disodium hydrogen orthophosphate (Na_2_HPO_4_·2H_2_O), sodium dihydrogen orthophosphate (NaH_2_PO_4_·2H_2_O),) and hydrochloric acid were purchased from Ajax Finechem Pty Ltd (Seven Hills, New South Wales, Australia). 4-Amino-6-hydroxy-2-mercaptopyrimidine monohydrate (AHMP), glucose, uric acid, albumin, and ascorbic acid (vitamin C) were purchased from Merck (Darmstadi, Germany). Artificial urine was purchased from Carolina Biological Supply Company (Burlington, NC, USA). Analytical grade reagents and 18 MΩ cm^−1^ resistance deionized (DI) water (obtained from a Millipore Milli-Q purification system) were used throughout. 

Phosphate buffers (pH 5–8) were prepared from 0.1 M of NaH_2_PO_4_·2H_2_O and 0.1 M Na_2_HPO_4_·2H_2_O. Phosphate buffer (pH 9–12) was prepared from 0.1 M of NaH_2_PO_4_·2H_2_O and 0.1 M Na_2_HPO_4_·2H_2_O and the pH was adjusted with 0.1 M sodium hydroxide (Merck) solution.

A wax printer (Xerox Color Qube 8570, Tokyo, Japan) was used to print wax onto transparency sheets to obtain the detection devices. Absorption spectra were measured by using a UV-2450 UV–visible spectrophotometer (SHIMADZU, Kyoto, Japan). The zeta potential measurements were performed on Zeta sizer Nano S90 (Malvern, UK). An atomic absorption spectrometer (AAS) with a hollow cathode lamp and standard air/acetylene flame was also used (Analyst 300, Perkin Elmer Instruments, Akron, OH, USA).

### 3.2. Preparation of AHMP-Modified Gold Nanoparticles (AHMP-AuNPs)

First, the purchased gold nanoparticles were preconcentrated by pipetting the commercial gold nanoparticles into 1 mL micro centrifuge tubes. The tubes were centrifuged at 12,000 rpm (4 °C) for 30 min. Then, 0.8 mL of supernatant was removed, and the remaining 0.2 mL of solution was stirred for 10 s.

AHMP-AuNPs were prepared by mixing 0.10 mL of 4 mM AHMP and 1.50 mL of the preconcentrated AuNP solution. After that, the mixture was stirred for 15 min to ensure the complete self-assembly of AHMP onto the surface of the AuNPs.

### 3.3. Fabrication of Transparency Sheet-Based Devices

Transparency sheet-based devices (polypropylene, purchased from local stationery store) were fabricated using a wax printing method. The pattern of the device was designed as an array containing a circular area with a detection zone 5 mm in diameter using the graphic software Adobe Illustrator CS4. The fabrication process has only two steps. First, the wax pattern was printed on the transparency sheet using the wax printer (Xerox Color Qube 8570, Japan). The maximum liquid volume in the detection zone was 30 μL. The wax-covered area was hydrophobic, while the detection area without wax was hydrophilic. Second, white paper was attached to the back side of the transparency sheet with tape to enhance the visibility of the color change. Using this process, the transparency sheet-based devices could be fabricated within 2 min. 

### 3.4. Transparency Sheet-Based Colorimetric Detection of Calcium

The working standard solution for the Ca^2+^ was made by appropriate dilution of the stock standard solution on a daily basis as required. A colorimetric analysis of Ca^2+^ was performed as follows. First, 5 µL of standard Ca^2+^ solution or samples (prepared in pH 6.0 PBS) were added into the detection zone of a transparency sheet-based device, followed by the addition of 25 µL of AHMP-AuNPs. Then, the red/blue intensity of AHMP-AuNPs mixed with Ca^2+^ standard solution was captured by mobile phone (Vivo V3, DongGuan, China) in a light control box and measured via Image J software (https://imagej.nih.gov). Under optimal conditions, the relationship between the red/blue intensity of triplicate measurements and concentrations of the Ca^2+^ solution between 10–100 ppm was plotted to obtain the linear range.

### 3.5. Analysis of Ca^2+^ in Artificial Urine Samples

Under optimal conditions, the proposed method was used for the determination of Ca^2+^ in artificial urine samples. The artificial urine was purchased from Carolina Biological Supply Company (controlled level). The proposed method was validated against atomic absorption spectrophotometry (AAS). An appropriate volume of 1000 ppm Ca^2+^ was added to 25 mL of artificial urine for preparation of the 500 ppm Ca^2+^ stock solution. Then, the stock solution was diluted with artificial urine to 320, 160, or 80 ppm Ca^2+^ concentration. The samples were adjusted to pH 6 by the addition of 3 M sodium hydroxide and additional urine was added to reach a final volume. Prior to analysis with the proposed method, the 320 and 160 ppm Ca^2+^ samples were diluted with 50 mM phosphate buffer pH 6, 4-fold and 2-fold, respectively. The AAS method can detect Ca^2+^ within the range of 0.5–5 ppm. Therefore, the 320, 160, and 80 ppm Ca^2+^ samples were diluted with 50 mM phosphate buffer pH 6, 150-fold, 45-fold, and 80-fold, respectively. 

## 4. Conclusions

This paper demonstrated, for the first time, the use of AHMP-AuNPs for a transparency sheet-based colorimetric method to provide simple, rapid, inexpensive, reliable, and portable devices for the determination of Ca^2+^ in urine samples. This method was based on the aggregation of AHMP-AuNPs on the device by the electrostatic force between the negatively charged AHMP-AuNPs and the positively charge Ca^2+^. Under optimal conditions, the color change of the AHMP-AuNPs can be clearly observed by the naked eye. Furthermore, the proposed method was successfully applied to the quantitative analysis of Ca^2+^ in artificial urine samples. This colorimetric device provided a result within one minute, much faster than other mentioned methods. This simple device is highly suitable for Ca^2+^ quantitative analysis and can act as an alternative method for the rapid screening of Ca^2+^ deficiency without the need for any special instrumentation. The advantages of the transparency sheet-based colorimetric device developed from this work include its high sensitivity and selectivity, good precision, simplicity, and reproducibility. These findings offer a high potential for applications in clinical diagnosis.

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
