# Peer review of "A Transparency Sheet-Based Colorimetric Device for Simple Determination of Calcium Ions Using Induced Aggregation of Modified Gold Nanoparticles"

_ijms, 2019, doi:10.3390/ijms20122954_

Round 1

Reviewer 1 Report

In this work, the authors reported the determination of Ca2+ using AHMP-functionalized AuNPs (AHMP-AuNPs) as a colorimetric indicator. However, this manuscript is really quite descriptive and many aspects of the manuscript need to be extensively improved. Most importantly, the main message, the proposed role of AHMP in the detection of Ca2+ has not been convincingly. In its current state, I cannot recommend publication of this manuscript on International Journal of Molecular Sciences.

There are at least five major deficiencies in the manuscript which need to be addressed before consideration for publication somewhere else. See below.

1. A key element of the proposed manuscript is establishing the interaction between AHMP-AuNPs and Ca2+ ions with high selectivity. However, in the current form, it is far from convincing. The authors stated “the electrostatic force between the AHMP-AuNPs and Ca2+ may induce aggregation (Line 89)” and “Ca2+ is known to bind easily with ligands containing lone pair electrons, such as –NH2 or –OH, via coordination bonds (Line 200)”. It was not obvious why other divalent ions could not induce AuNP aggregation by AHMP-AuNPs. It will be indispensable to provide stronger evidences and references for your hypothesis.

2. In Figure S1, the authors explained the red/blue intensity of the blank was increased due to the evaporation effect which does not make sense. If the R/B intensity would be affected by the evaporation, all conditions (the addition of Ca2+ and Mg2+) should have the same trend. Why only the evaporation could be observed in the blank solution?

3. The selectivity of AHMP-AuNPs toward Ca2+ is less clear. For instance, in Figure S1, AHMP-AuNPs with Mg2+ and the blank have the almost same R/B intensity value which means that they have the same color in red. In apparent contradiction, Figure 5 showed the different color; the blank was in red and AHMP-AuNPs with Mg2+ was in purple. In addition, Mg2+ seems to have a certain degree of effect on the selectivity of AHMP-AuNPs. The authors should discuss this part and provide the solution.

4. Following Question 3 and 1, the results of selectivity for other divalent ions should be included in Figure 5. Also, please define the ΔR/B intensity.

5. Another apparent contradiction was found in Figure 6. The R/B intensity of blank was less than 3 in Figure 3, 4, S1 and S2. Why the R/B intensity of the Ca2+ concentrations ranging from 10~30 ppm were beyond the value of 3 in Figure 6?

Reviewer 2 Report

The present manuscript ‘A Transparency Sheet-Based Colorimetric Device for Simple Determination of Calcium Ions using Induced Aggregation of Modified Gold Nanoparticles’ studies calcium detection by Transparency Sheet-Based Colorimetric Device. These findings might offer a high potential for applications in clinical diagnosis. The study looks good and may be accepted after addressing the following comments;

1.     The abstract should be revised carefully; the author added more introductory part.

2.     Add a real photograph of solution colors to Scheme 1.

3.     Fig. 3 and 4 bars should be labeled in the figure

4.     Diagrammatic representation of the Transparency Sheet-Based Colorimetric Device is required.

Author Response

Reviewer II

Comments and Suggestions for Authors

The present manuscript ‘A Transparency Sheet-Based Colorimetric Device for Simple Determination of Calcium Ions using Induced Aggregation of Modified Gold Nanoparticles’ studies calcium detection by Transparency Sheet-Based Colorimetric Device. These findings might offer a high potential for applications in clinical diagnosis. The study looks good and may be accepted after addressing the following comments;

1.     The abstract should be revised carefully; the author added more introductory part.

For this suggestion, the abstract was revised and more literature reviews were added in the introduction as highlighted.

2.     Add a real photograph of solution colors to Scheme 1.

We have added real photograph of solution colors to Scheme 1 following reviewer’s suggestion.

3.     Fig. 3 and 4 bars should be labeled in the figure

We have added labels in figure 3 and 4 following reviewer’s suggestion.

4.     Diagrammatic representation of the Transparency Sheet-Based Colorimetric Device is required.

            We have added diagrammatic representation of the Transparency Sheet-Based Colorimetric Device in Scheme 1

Reviewer 3 Report

The authors have described a colourimetric method for detection and determination of Ca ion based on transparency sheets system using modified gold nanoparticles. I highly enjoyed the reading of the article and recommend publishing it in the International Journal of Molecular Sciences. However, a few points have to be considered first:

1- In section 3.4: The authors have not provided a rationale for the high selectivity of their method for Ca detection compared to other different divalent cations although they have mentioned that detection based on simple electrostatic interaction between negatively charged AHMP-AuNPs and positively charged calcium ions.

2- In section 3.1: authors have reported that zeta potential of AuNPs after their modification is found to be 49.9mV whilst in section 3.2 the zeta potential for the same AuNPs platform was given as 58.2 mv, so consistency should have adhered.

3- I recommend adding more examples as references for use of AuNPs as colourimetric probes

- Talanta, 2017, 169, 13-19

- Arabian Journal of Chemistry, 2018, 11, 1134-43

Author Response

Reviewer III

Comments and Suggestions for Authors

The authors have described a colorimetric method for detection and determination of Ca ion based on transparency sheets system using modified gold nanoparticles. I highly enjoyed the reading of the article and recommend publishing it in the International Journal of Molecular Sciences. However, a few points have to be considered first:

1- In section 3.4: The authors have not provided a rationale for the high selectivity of their method for Ca detection compared to other different divalent cations although they have mentioned that detection based on simple electrostatic interaction between negatively charged AHMP-AuNPs and positively charged calcium ions.

            Thank you for this suggestion. To be clearer, we have newly written and emphasized the effect of incubation time used for doing the experiment to reader again in section of selectivity as highlighted in text page 8 line 270-283.

2- In section 3.1: authors have reported that zeta potential of AuNPs after their modification is found to be 49.9mV whilst in section 3.2 the zeta potential for the same AuNPs platform was given as 58.2 mv, so consistency should have adhered.

Thank you very much for your comment.  The zeta potential of modified AuNPs changed from -49.9 mV (in section 3.1) to -58.2 mv (in section 3.2) due to the addition of buffer into modified AuNPs.  Therefore, the negative charge of solution increased from -49.9 mV to -58.2 mV. To be clearer, we have rewritten in text as highlighted with yellow.

3- I recommend adding more examples as references for use of AuNPs as colorimetric probes

- Talanta, 2017, 169, 13-19

- Arabian Journal of Chemistry, 2018, 11, 1134-43

For this comment, we have added suggested references in introduction part as highlighted.

Round 2

Reviewer 1 Report

The authors tried to improve the original ones, but the results were not satisfactory. I suggest refocusing the manuscript in terms of verifying data consistency.

1 Line 205: The authors stated that “Ca2+ is known to bind easily with ligands containing lone pair electrons, such as –NH2 or –OH, via coordination bonds”. Do the authors have references to add here in order to support their statement?

2. In Figure S1, the R/B intensity of blank in 1 minute is ~2.5 at the volume ratio (AHMP-AuNP:sample) of 1 : 1. However, in the following experiment (Figure S2), the R/B intensity of blank in 1 minute is ~1.7 at the same volume ratio (15:15). I also find the R/B intensities of blank in all volume ratio of Figure S2 are lower than that of Figure S1 in 1-minute incubation. This appears to need the author’s explanation. If there have different conditions, please describe them in the figure caption in detail.

3. In section 3.4, why the authors replace " Red/Blue intensity” with "Δ Red/Blue intensity"? The authors should give the definition of “Δ Red/Blue intensity” at the first time when this term is used in section 3.4.

4. Line 281: The authors stated that “This might be Ca2+ has the ionic radius appropriated with size selective of AHMP-AuNPs after aggregation. “ Do the authors have evidence to support their statement? If no, please delete this description.

5. In Figure 5, the error bars should be shown.

6. Line 277: …AuNPs mixed with foreign compounds with concentrations even 10 times greater than the concentration of Ca2+ did not significantly affect the signal…

The caption of Figure 5: Responses of AHMP-AuNPs in the other interferences as 100 ppm.

I was left confused the concentration information for Ca2+ and other interference ions. Thus, what are the concentrations of other interference ions? If 100 ppm, according to the description of line 277, the concentration of Ca2+ should be 10 ppm?

7. In Figure 5, the color of AHMP-AuNP solution with Mg2+ looks like purple (Δ R/B intensity = 0.1) which is different from those with other interference ions in red. In Figure 6, the purple color of AHMP-AuNP solution can be observed in the addition of 60 ppm Ca2+, at this time, Δ R/B intensity is higher than 2. Taken together with these two figures, why the almost same color is corresponding to significantly different values of Δ R/B intensity???

8. The same situation can be seen from the AHMP-AuNP solution with 100 ppm Ca2+ in Figure 5 & 6. In Figure 5, Δ R/B intensity of 100 ppm Ca2+ is ~2.5 based on the three experiments (n=3). Why the Δ R/B intensity of 100 ppm Ca2+ is greater than 3 in Figure 6?

It is apparent that there are other contradictions between selectivity and sensitivity results in the figure 5& 6, and the description of section 3.4 & 3.5. Thus, please thoughtfully revise these two sections and two figures.

Author Response

Reply to reviewers comment

            First of all, we would like to thank reviewer for his evaluations. The comments and suggestions are very useful for us to improve our manuscript. We do appreciate for this matter.  The answers of each comment are shown below subsequently.

Comments and Suggestions for Authors

The authors tried to improve the original ones, but the results were not satisfactory. I suggest refocusing the manuscript in terms of verifying data consistency.

1 Line 205: The authors stated that “Ca2+ is known to bind easily with ligands containing lone pair electrons, such as –NH2 or –OH, via coordination bonds”. Do the authors have references to add here in order to support their statement?

Thank you very much for this comment. We have read references and rewritten new sentence as highlighted in line 205-206.  In addition, we have added references [31, 32] in manuscript.

2. In Figure S1, the R/B intensity of blank in 1 minute is ~2.5 at the volume ratio (AHMP-AuNP:sample) of 1 : 1. However, in the following experiment (Figure S2), the R/B intensity of blank in 1 minute is ~1.7 at the same volume ratio (15:15). I also find the R/B intensities of blank in all volume ratio of Figure S2 are lower than that of Figure S1 in 1-minute incubation. This appears to need the author’s explanation. If there have different conditions, please describe them in the figure caption in detail.

 As reviewer commented that R/B intensity values obtained from Figure S1 and S2 are different.  To be honest, Figure S1 is the preliminary results.  At that time, we used different controlling light box for capture images.  Therefore, the intensity is quite different.   Moreover, the results obtained from Figure S2 used all optimal conditions.   To be clearer, figure captions of Figure S1 and Figure S2 were reported clearly.

3. In section 3.4, why the authors replace "Red/Blue intensity” with "Δ Red/Blue intensity"? The authors should give the definition of “Δ Red/Blue intensity” at the first time when this term is used in section 3.4.

First, red/blue intensity was used to evaluate the optimal conditions obtained from each experimental parameter.  After getting the optimal conditions, the quantitative analysis and selectivity was evaluated by using Dred/blue intensity because we believed that Dred/blue intensity provided higher accuracy than red/blue intensity.   In addition, we have added the definition of Δ red/blue intensity (red/blue intensity of blank – red/blue intensity of sample) in manuscript line 270 as highlighted.

4. Line 281: The authors stated that “This might be Ca2+ has the ionic radius appropriated with size selective of AHMP-AuNPs after aggregation. “ Do the authors have evidence to support their statement? If no, please delete this description.

For this suggestion, we have deleted unclear sentence from manuscript.

5. In Figure 5, the error bars should be shown.

 We have added the error bars in figure 5 following reviewer’s suggestion.

6. Line 277: …AuNPs mixed with foreign compounds with concentrations even 10 times greater than the concentration of Ca2+ did not significantly affect the signal…

The caption of Figure 5: Responses of AHMP-AuNPs in the other interferences as 100 ppm.

I was left confused the concentration information for Ca2+ and other interference ions. Thus, what are the concentrations of other interference ions? If 100 ppm, according to the description of line 277, the concentration of Ca2+ should be 10 ppm?

We are sorry for our mistake. We have rewritten the figure caption of Figure 5 as highlighted.

7. In Figure 5, the color of AHMP-AuNP solution with Mg2+ looks like purple (Δ R/B intensity = 0.1) which is different from those with other interference ions in red.  In Figure 6, the purple color of AHMP-AuNP solution can be observed in the addition of 60 ppm Ca2+, at this time, Δ R/B intensity is higher than 2. Taken together with these two figures, why the almost same color is corresponding to significantly different values of Δ R/B intensity?

Thank you very much for this comment.  We are sorry for this mistake. We have corrected Figure 5.  The data in new figure 5 was presented using the optimal conditions as same as figure 6.

8. The same situation can be seen from the AHMP-AuNP solution with 100 ppm Ca2+ in Figure 5 & 6. In Figure 5, Δ R/B intensity of 100 ppm Ca2+ is ~2.5 based on the three experiments (n=3). Why the Δ R/B intensity of 100 ppm Ca2+ is greater than 3 in Figure 6?

It is apparent that there are other contradictions between selectivity and sensitivity results in the figure 5& 6, and the description of section 3.4 & 3.5. Thus, please thoughtfully revise these two sections and two figures.

From reviewer’s comments.  We have revised Figure 5 by using all conditions as same as figure 6.

Round 3

Reviewer 1 Report

I thank the authors for addressing my comments. I have no further comments regarding data presentation and interpretation.